# OpenReview forum: "Sparse Models, Sparse Safety: Unsafe Routes in Mixture-of-Experts LLMs"
_ICLR.cc/2026/Conference — Submitted to ICLR 2026_

### Official Review · Reviewer_2yeh · 2025-10-27

**Soundness:** 3
**Presentation:** 2
**Contribution:** 3
**Rating:** 6
**Confidence:** 3

**Summary:**

This paper studies the safety of MoE LLMs and discovers unsafe routes within the sparse architecture. The authors first propose RoSais to quantify the safety importance of layers across layers. This is achieved by randomly masking routing decision and observing the model output changes. Then, the authors find that manipulating the safety-critical routers, the ones with higher RoSais, cause the models to generate unsafe content. Furthermore, the authors propose a more effective attack framework F-SOUR to achieve an average ASR of 0.9, considering the interaction between shallow and deep routers after intervention. Defensive perspectives are also discussed.

**Strengths:**

1. **Good research question**: The paper addresses an important and underexplored problem — the safety of routing mechanisms in MoE large language models. While prior work has primarily focused on improving utility and efficiency, this study fills a crucial gap by systematically examining router safety.
2. **Intuitive approach to find safety-critical routers**: The intuition behind identifying safety-critical routers is clear and insightful: the importance can be revealed by observing how the model’s output changes when its routing decisions are manipulated.
3. **Extensive experiments**: The authors evaluate four models, compare against four baseline approaches, and propose two methods (RoSais-based unsafe route discovery and F-SOUR), complemented by a case study on the defense mechanism. This strengthens the validity of the findings.

**Weaknesses:**

1. The main concern lies in the trivial nature of the central finding regarding safety-critical routers. Since routers are designed to assign expert FFNs based on the characteristics of the input task, it is expected that risky or harmful inputs would naturally be routed to specific experts. Thus, identifying these “safety-critical routers” may reflect an inherent property of MoE routing rather than a novel safety insight.
2. RoSais is based on the model's next-token distribution given a harmful question (line 197). However, prior work [1] suggests that next-token distributions are insufficient as reliable indicators of model safety. This raises concerns about whether RoSais accurately captures safety-relevant behavior and how this limitation may affect its reliability.
3. The approach to find and leverage safety-critical routers depends on a random sampling mechanism (line 186, 248). However, the paper does not provide details about the computational cost of this procedure, which could limit its practicality for large-scale MoE models.

[1] Safety alignment should be made more than just a few tokens deep

**Questions:**

1. Do the identified safety-critical routers generalize across different harmful questions or categories of unsafe content? For example, if a router is found critical for one harmful prompt (Q1), does it remain critical for another (Q2)?
2. Given that prior studies [1] suggest next-token distributions may not fully reflect model safety, how do the authors justify using this signal as the basis for RoSais? Would incorporating multi-token contexts improve robustness?
3. What is the computational overhead of the random sampling approach (line 186, 248) used to identify and leverage safety-critical routers? Could this process scale to larger MoE models?

[1] Safety alignment should be made more than just a few tokens deep

---

> ### Author Response · Authors · 2025-11-21
> **Response to Reviewer 2yeh (1/2)**
>
> Thank you very much for reviewing our work and acknowledging its good research question, intuitive approach, and extensive experiments. We note that your main concerns relate to our central finding, token-depth, and computational overhead. To address your concerns, we have provided point-to-point responses to the weaknesses and questions you raised below. Besides, we have carefully revised our manuscript and highlighted the changes in blue.
>
> - W1: *The main concern lies in the trivial nature of the central finding regarding safety-critical routers. Since routers are designed to assign expert FFNs based on the characteristics of the input task, it is expected that risky or harmful inputs would naturally be routed to specific experts. Thus, identifying these “safety-critical routers” may reflect an inherent property of MoE routing rather than a novel safety insight.*
>     - Thank you for your insightful comment. Indeed, some previous work [1-3] demonstrates the existence of domain experts. For instance, some experts are preferred for paper writing or code generation. However, unsafe tasks can span different domains, such as “how to make a bomb” and “how to write a virus program,” resulting in different expert-selection routes. Hence, different risky or harmful inputs would produce different routes.
>     - At the dataset-level RoSais-based, we attempt to find an unsafe route that is effective for harmful questions across different domains and achieves a non-trivial performance. However, this is insufficient. Because each token of the input question would trigger each router to produce different routing scores, MoE LLM’s routes are highly dynamic and specialized. Hence, we propose F-SOUR, which accurately finds a specialized unsafe route for any given harmful question across all evaluated models and datasets, far exceeding the evaluated baselines and our RoSais-based attacks.
>     - In summary, the attack surface brought by routers is inherent to MoE LLMs, but our findings extend beyond static safety-critical routers to more fine-grained unsafe routes.
> - W2&Q2: *Given that prior studies [1] suggest next-token distributions may not fully reflect model safety, how do the authors justify using this signal as the basis for RoSais? Would incorporating multi-token contexts improve robustness?*
>     - Thank you for the valuable suggestion. To validate the effectiveness of our proposed RoSais score, we further conduct an ablation to compare the performance of our proposed RoSais-based methods and random masking on DeepSeek-V2-Lite. Specifically, for any given harmful question, random masking would randomly select *L* = {1, 2, 5} layers and add a random mask to the original routing score for each selected layer. The evaluation results (ASR) are shown in the table below. We notice that our methods consistently outperform the random masking baseline and can significantly boost the ASR by up to 0.64 (JailbreakBench) and 0.88 (AdvBench), while the random masking baseline can only increase the ASR up to 0.09 (JailbreakBench) and 0.18 (AdvBench). This empirically demonstrates that the ASR improvements of RoSais-based attacks are primarily due to our RoSais-based routing manipulations, rather than random masks. We have added this ablation to Appendix C to better explain the effectiveness of the RoSais score.
>
>         | Dataset | \# Changed Layers | Sample-Level RoSais | Dataset-Level RoSais | Random Masking |
>         | --- | --- | --- | --- | --- |
>         | JailbreakBench | 0 | 0.15 | 0.15 | 0.15 |
>         | JailbreakBench | 1 | 0.50 | 0.27 | 0.24 |
>         | JailbreakBench | 2 | 0.45 | 0.53 | 0.21 |
>         | JailbreakBench | 5 | 0.46 | 0.79 | 0.24 |
>         | AdvBench | 0 | 0.02 | 0.02 | 0.02 |
>         | AdvBench | 1 | 0.48 | 0.34 | 0.20 |
>         | AdvBench | 2 | 0.52 | 0.64 | 0.16 |
>         | AdvBench | 5 | 0.56 | 0.90 | 0.14 |
>
>     - We only consider the next token in RoSais because it has less computational overhead. As can be seen from the ASR of RoSais-based attacks, simply considering the next token can significantly reduce the safety of MoE LLMs. Regarding multi-token contexts, we use this setting in F-SOUR, that is, optimizing a corresponding harmful target for a given harmful question and achieving a performance far superior to RoSais-based attacks (i.e., first-token-only). To better motivate our F-SOUR design, we emphasize the limitations of RoSais, only considering the next token in Section 4, and cite [4] to explain the rationale for our F-SOUR to consider multiple tokens.

---

> ### Author Response · Authors · 2025-11-21
> **Response to Reviewer 2yeh (2/2)**
>
> - W3&Q3: *What is the computational overhead of the random sampling approach (line 186, 248) used to identify and leverage safety-critical routers? Could this process scale to larger MoE models?*
>     - The computation of RoSais is only related to the model size in terms of the number of MoE layers (i.e., the number of routers). Specifically, during RoSais estimation, given $|\mathbb{Q}|$ harmful questions and an MoE LLM with $L$ MoE layers, we perform $S_1$ random masking operations on each layer. Then the overall computation count is $|\mathbb{Q}| \cdot L \cdot S_1$. For RoSais-based attacks, replacing $S_1$ and $L$ with $S_2$ and $L_\Phi$, the overall computation count becomes $|\mathbb{Q}| \cdot L_\Phi \cdot S_2$.
>     - The main computation lies in prefilling and decoding the first token (like an inference that only needs to output a token). Besides, the additional memory required for the random mask is negligible (only a few tens of bytes). In this work, due to resource constraints, we consider different MoE LLMs ranging from 16 (i.e., OLMoE-1B-7B with 7B parms) to 32 MoE layers (i.e., Mixtral-8x7B with 56B params). With sufficient GPU memory, our method can be extended to larger MoE LLMs (e.g., DeepSeek-V3 with 58 MoE layers) without causing a huge increase in computation count, as it merely increases linearly with the number of MoE layers.
> - Q1: *Do the identified safety-critical routers generalize across different harmful questions or categories of unsafe content? For example, if a router is found critical for one harmful prompt (Q1), does it remain critical for another (Q2)?*
>     - Thank you for your helpful suggestion. Our dataset-level RoSais-based attack results (Table 1) show that there exists an unsafe route that can be consistently applied to different harmful questions. This means that a router could be critical for both Q1 and Q2. Furthermore, as shown in Figures 3a and 6a, the highest-RoSais routers concentrate on a few layers rather than being uniformly distributed, indicating that at the sample-level, some harmful questions share the same critical routers.
>     - Furthermore, to evaluate the transferability of the unsafe routes obtained by our proposed dataset-level RoSais-based attack, we further conduct a cross-dataset evaluation. Specifically, in the cross-dataset setting, if the dataset being evaluated is JailbreakBench, then the dataset-level unsafe route is obtained from another dataset (i.e., AdvBench), and vice versa.
>     - The table below shows the ASR of our proposed dataset-level RoSais-based attacks considering cross-dataset transferability on DeepSeek-V2-Lite, where the values in parentheses represent the differences caused by the cross-dataset setting. We notice that unsafe routes have strong transferability among different datasets. The cross-dataset ASR decreases by less than 0.10 in most cases, with minimum and maximum decreases of 0.00 and 0.16, respectively. For the best-performing case of changing 5 layers, the transferred unsafe routes achieve ASRs of 0.69 and 0.86 on JailbreakBench and AdvBench, respectively.
>     - We have incorporated this result into Appendix C of the revision to demonstrate the transferability of our dataset-level RoSais-based attack.
>
>         | Dataset | # Changed Layers | Is Cross-Dataset? | ASR |
>         | --- | --- | --- | --- |
>         | JailbreakBench | 0 | N/A | 0.15 |
>         | JailbreakBench | 1 | No | 0.27 |
>         | JailbreakBench | 1 | Yes | 0.27 (±0.00) |
>         | JailbreakBench | 2 | No | 0.53 |
>         | JailbreakBench | 2 | Yes | 0.46 (-0.07) |
>         | JailbreakBench | 5 | No | 0.79 |
>         | JailbreakBench | 5 | Yes | 0.69 (-0.10) |
>         | AdvBench | 0 | N/A | 0.02 |
>         | AdvBench | 1 | No | 0.34 |
>         | AdvBench | 1 | Yes | 0.32 (-0.02) |
>         | AdvBench | 2 | No | 0.64 |
>         | AdvBench | 2 | Yes | 0.48 (-0.16) |
>         | AdvBench | 5 | No | 0.90 |
>         | AdvBench | 6 | Yes | 0.86 (-0.04) |
>
> [1] Qingyue Wang, Qi Pang, Xixun Lin, Shuai Wang, and Daoyuan Wu. BadMoE: Backdooring Mixture-of-Experts LLMs via Optimizing Routing Triggers and Infecting Dormant Experts. CoRR abs/2504.18598, 2025.
>
> [2] Yuanbo Tang, Yan Tang, Naifan Zhang, Meixuan Chen, and Yang Li. Unveiling Hidden Collaboration within Mixture-of-Experts in Large Language Models. CoRR abs/2504.12359, 2025.
>
> [3] Haomin Zhuang, Yihua Zhang, Kehan Guo, Jinghan Jia, Gaowen Liu, Sijia Liu, and Xiangliang Zhang. SEUF: Is Unlearning One Expert Enough for Mixture-of-Experts LLMs? In Annual Meeting of the Association for Computational Linguistics (ACL), pp. 8664–8678. ACL, 2025.
>
> [4] Xiangyu Qi, Ashwinee Panda, Kaifeng Lyu, Xiao Ma, Subhrajit Roy, Ahmad Beirami, Prateek Mittal, and Peter Henderson. Safety Alignment Should Be Made More Than Just a Few Tokens Deep. In International Conference on Learning Representations (ICLR), 2025.

---

> ### Author Response · Authors · 2025-11-27
>
> Dear Reviewer 2yeh,
>
> Thank you very much for your thoughtful review of our submission. During the rebuttal, we have provided explanations and revised our manuscript carefully based on your insightful suggestions.
>
> We care deeply about whether our rebuttal has addressed your concerns. If so, we'd greatly appreciate it if you could consider re-evaluating our submission. If you require any additional clarification to support your re-evaluation, we are committed to responding promptly and cooperating fully to address any remaining questions.
>
> Thank you again for your dedicated time and invaluable contributions.

---

> ### Comment · Reviewer_2yeh · 2025-11-28
> **Feedback to the authors' response**
>
> Greetings the authors,
>
> Thank you for your response to my concerns. Below is the feedback:
>
> **W1.** I have no more question on this, thanks for your clarification.
>
> **W2&Q2.** Thank you for your response! However, I don't understand why you provide comparison to a baseline, random masking, to illustrate the potential limitation of next-token distribution. I am concerned that there are cases where the unsafe response doesn't appear at the first token. For example:
> > Q: Tell me how to make the bomb.
> > A: To make a bomb, you need to...
>
> The first token "To" is not harmful and is not in your affirmative token list in Table 3.
>
> **W3&Q3.** Thank you for providing the equation! However, it appears to be the computation count, rather than the actual runtime or computational complexity. For example, for a specific harmful question, how long does it take in practice to obtain the unsafe routes? And how does this runtime scale as the MoE model size increases?
>
> **Q1.** Thank you for your response! However, there appears to be a contradiction between the statements that "each token of the input question would trigger each router to produce different routing scores" (response for W1) and that "unsafe routes have strong transferability among different datasets."
>
> Since my concerns are not fully addressed, I will maintain my score.

---

> > ### Author Response · Authors · 2025-11-29
> > **Response to Reviewer 2yeh's Feedback (1/2)**
> >
> > Dear Reviewer 2yeh,
> >
> > Thank you very much for your feedback. We provide answers below to further address your concerns.
> >
> > - W2&Q2: *However, I don't understand why you provide comparison to a baseline, random masking, to illustrate the potential limitation of next-token distribution.*
> >     - Thank you for checking our experiment on random masking. We provide this experiment to verify the effectiveness of our proposed RoSais and to illustrate "W2: RoSais accurately captures safety-relevant behavior." Specifically, since RoSais significantly outperforms random masking, it demonstrates that compared to randomly masking routers, RoSais captures safety-relevant behavior by increasing the probability of outputting affirmative tokens.
> >
> > - W2&Q2: *I am concerned that there are cases where the unsafe response doesn't appear at the first token.*
> >     - Thank you for providing this example. Admittedly, the affirmative tokens we use in Table 3 (or Table 4 in the revision) cannot cover all possible first tokens for unsafe responses. The Limitation and Future Work section acknowledges exploring a wider range of targets as a worthwhile future direction.
> >     - Furthermore, by having the MoE LLM output complete answer tokens instead of just the first token, the adversary can further use a judge model (e.g., StrongReject) to calculate an unsafe score as a more reliable judgment metric. However, this approach introduces a significantly higher computational cost than current methods. For instance, on DeepSeek-V2-Lite, the average number of input tokens in JailbreakBench is 22.37. If the evaluation length of the complete answer is 128, then even without considering the calculation of the judge model, the computational cost on the current MoE LLM is approximately $\frac{(22.37 + 128)}{(22.37 + 1)} \approx 6.43\times$ for generating only the first token. Even if the RoSias-based attack only considers the first token, it still achieves an average ASR of 0.59 (better than other existing baselines). **For efficiency and cost-effectiveness, RoSias simplifies this by only considering the first token. For better attack performance, we propose F-SOUR, which considers multiple tokens and achieves an average ASR of 0.90.**

---

> > ### Author Response · Authors · 2025-11-29
> > **Response to Reviewer 2yeh's Feedback (2/2)**
> >
> > - W3&Q3: *However, it appears to be the computation count, rather than the actual runtime or computational complexity.*
> >     - Thank you for your feedback. For a harmful question containing $N$ tokens, assume the number of activated parameters and routed layers of the target MoE LLM are $P_{Act}$ and $L_{Routed}$, respectively. The RoSais-based attack obtains the unsafe route for the harmful question via two steps: Step I. computing RoSais for each routed layer, and Step II. choosing a mask for high-RoSais layer(s). We provide the computational complexity of each step below.
> >     - For Step I, a total of $S_1 \cdot L_{Routed}$ random masks need to be selected ($S_1$ is a pre-defined constant), $N$ tokens are prefilled, and the first token is decoded. $P_{Act}$ represents the number of parameters activated in the MoE LLM for each token. Therefore, the computational complexity of this step is $\mathcal{O}(P_{Act} \cdot S_1 \cdot L_{Routed} \cdot (N + 1))$. For Step II, a total of $S_2 \cdot L_\Phi$ random masks need to be selected ($S_2$ is a pre-defined constant), $N$ tokens are prefilled, and the first token is decoded. Hence, the computational complexity of this step is also $\mathcal{O}(P_{Act} \cdot S_2 \cdot L_\Phi \cdot (N + 1))$. Overall, the computational complexity of the RoSais-based attack is $\mathcal{O}(P_{Act} \cdot (S_1 \cdot L_{Routed} + S_2 \cdot L_\Phi) \cdot (N + 1))$. Since $S_1$, $S_2$, $L_\Phi$, and $N$ are generally independent of the model size, our computational complexity is primarily linearly related to $P_{Act} \cdot L_{Routed}$ as the model size increases. Compared to the largest model we evaluated (Mixtral-8x7B), a larger MoE LLM, DeepSeek-V3's computational complexity increases by approximately $4 \times$, which indicates our method could be scaled to large models.
> >     - The table below lists the average time spent by different MoE LLMs on each harmful question using the NVIDIA A100-SXM4-80GB on JailbreakBench. To illustrate the results, we use the default settings: $S_1=20$, $S_2=100$, and $L_\Phi = 1$. Because different model may use different tokenizers and chat templates, their average number of input tokens varies. We observe that as the model size increases, the attack time increases (similar to the computational complexity). However, due to factors such as model architecture affecting the actual inference latency [1], the percentage increase in time consumption is not necessarily consistent with the percentage increase in computational complexity.
> >     - **Overall, these results demonstrate that our method has the potential to be applied to larger models.**
> >
> >         | Model | Actived Params (B) | # Routed Layers | Average # Input Tokens | Average Time (Seconds) |
> >         | --- | --- | --- | --- | --- |
> >         | DeepSeek-V2-Lite | 2.4 | 25 | 22.37 | 99.84 |
> >         | Mixtral-8x7B | 13 | 32 | 25.39 | 528.21 |
> >         | OLMoE-1B-7B | 1 | 16 | 28.72 | 94.50 |
> >         | Qwen1.5-MoE-A2.7B | 2.7 | 24 | 34.15 | 190.59 |
> >
> > - Q1: *However, there appears to be a contradiction between the statements that "each token of the input question would trigger each router to produce different routing scores" (response for W1) and that "unsafe routes have strong transferability among different datasets."*
> >     - Thank you for noticing this. The two statements are not contradictory because they refer to different attacks. In our RoSais-based attack, we apply unsafe routes directly to MoE LLMs (Eq. 11), consistently acting on each input token. Therefore, we can find some transferable unsafe routes and discover that "unsafe routes have strong transferability among different datasets." However, since in real-world applications, "each token of the input question would trigger each router to produce different routing scores," RoSais-based attacks omit token-wise routing dynamics (L365-L366 in the original submission, L374-375 in the revision). Therefore, we propose a fine-grained attack (F-SOUR) to capture these dynamics and achieve better performance.
> >
> > Although we are unable to receive further responses from you due to rule restrictions, we sincerely hope that our response could resolve your concerns about our work.
> >
> > [1] Song Bian, Minghao Yan, and Shivaram Venkataraman. Scaling Inference-Efficient Language Models. CoRR abs/2501.18107, 2025.

---

### Official Review · Reviewer_H7YM · 2025-10-31

**Soundness:** 3
**Presentation:** 3
**Contribution:** 2
**Rating:** 2
**Confidence:** 4

**Summary:**

This paper reveals that the safety of Mixture-of-Experts (MoE) large language models is as sparse as their architecture; manipulating only a few critical routers can convert safe outputs into harmful ones. The paper introduces the Router Safety Importance Score (RoSais) and a fine-grained optimization method (F-SOUR) to identify unsafe routing configurations, achieving high attack success rates and highlighting the need for safety-aware router defenses.

**Strengths:**

1. The proposed framework enables the discovery of unsafe routing paths with token-level precision, highlighting a sophisticated methodological design.

2. Through controlled experiments across multiple MoE LLM families, the paper convincingly demonstrates that manipulating only a few routers can dramatically increase attack success rates, providing strong empirical validation of the threat.

**Weaknesses:**

1. Incremental contribution. The contribution appears limited and incremental, given concurrent efforts exploring similar safety issues in MoE LLMs. Previous works [1,2] have already shown that altering routers can induce harmful outputs. This paper primarily introduces a router scoring mechanism similar to a filtering approach, which may not represent a substantial conceptual advance.

2. Lack of comparison with recent baselines. The paper omits comparisons with concurrent works [1,2], both of which analyze router-related vulnerabilities and jailbreak effects in MoE LLMs. Without such baselines, it is difficult to assess the novelty and relative effectiveness of the proposed method.

3. Unclear effectiveness of the RoSais score. The improvement reported in Table 1 may stem from random masking rather than genuine routing manipulation. The paper lacks a solid ablation or sensitivity analysis to isolate the effect of RoSais.

[1] Mohsen Fayyaz, Steering MoE LLMs via Expert (De)Activation.
[2] Zhenglin Lai, SAFEx: Analyzing Vulnerabilities of MoE-based LLMs via Stable Safety-critical Expert Identification.

**Questions:**

1. When a specific router is modified, subsequent routers may also change their activation patterns. How does the paper ensure that the proposed metric accurately quantifies safety criticality without being confounded by downstream routing effects?

---

> ### Author Response · Authors · 2025-11-21
> **Response to Reviewer H7YM (1/2)**
>
> Thank you very much for reviewing our work and acknowledging our sophisticated methodological design and strong empirical validation. We note that your main concerns relate to the comparison with concurrent work and the effectiveness of RoSais score. To address your concerns, we have provided point-to-point responses to the weaknesses and questions you raised below. Besides, we have carefully revised our manuscript and highlighted the changes in blue.
>
> - W1&W2: *Incremental contribution & Lack of comparison with recent baselines*
>     - Thank you for your careful comment. **However, according to the ICLR 2026 Reviewer Guide (https://iclr.cc/Conferences/2026/ReviewerGuide), we do not need to compare works published within the last two months or those only on arXiv.** For your reference, we have put the reviewer guide’s explanation regarding very recent work below.
>
>         ```
>         We consider papers contemporaneous if they are published within the last two months. That means, since our full paper deadline is September 24, if a paper was published (i.e., at a peer-reviewed venue) on or after July 24, 2025, authors are not required to compare their own work to that paper. Note that arXiv is not considered a peer-reviewed venue. As such, authors are not required to compare to papers solely on arXiv: they may be excused for not knowing about papers not published in peer-reviewed conference proceedings or journals, which includes papers exclusively available on arXiv.
>
>         While authors are not required to compare to contemporaneous work or unpublished arxiv papers, they are strongly encouraged to cite such related work if they are aware of it. Reviewers can make authors aware of related contemporaneous work or arxiv papers, but the lack of such comparisons cannot be a basis for rejection.
>         ```
>
>     - Specifically, SteerMoE [1] was published on arXiv on September 11, 2025, while SAFEx [2] was published on arXiv on June 20, 2025, and is expected to be published in the formal peer-reviewed venue (NeurIPS’25) in December 2025. **Therefore, we do not need to compare these two papers.** Even so, we have already introduced these two concurrent works in Section 2.2 and technically compared our differences with theirs. Specifically, concurrent work primarily focuses on discovering the existence of safety experts and diagnosing activation differences. However, how serious the safety threats brought by safety experts can be, especially under worst-case scenarios, has never been well quantified and studied. Because their methods mainly observe (analyze) rather than search, they typically result in slight increases in harmful outputs. In contrast, we take a proactive, route-based approach that (i) searches over routing selections rather than only diagnostic activation differences and (ii) optimizes cross-token routing sequences (not just expert sets). Our later experimental results also demonstrate that their proposed methods for safety-critical experts are not essential, as their manipulation of the candidate experts cannot totally convert the model's safety (increasing ASR by only ~0.20). In contrast, our exploration on "unsafe route" can successfully convert the model's safety significantly (increasing ASR by over 0.70), demonstrating the fundamental exploration and weaknesses in MoE safety.
>     - In addition, we note that by the full paper deadline of ICLR 2026 (September 24, 2025), SteerMoE has released a publicly available code repository (containing usable steering vectors), while SAFEx has not. Hence, to better demonstrate the superiority of our proposed method over concurrent work, we further compare the performance (ASR) of our proposed F-SOUR and SteerMoE on two MoE LLMs (SteerMoE does not provide steering vectors or scripts for reproducing the steering vectors for the other two MoE LLMs). The results (ASR) are demonstrated in the table below. We observe that F-SOUR consistently outperforms SteerMoE on all evaluated LLMs and datasets. Although SteerMoE achieves attack performance comparable to other attacks (such as GCG and SHIPS), the average ASR of SteerMoE (e.g., 0.44 on AdvBench) is even lower than half of that of F-SOUR (e.g., 0.98 on AdvBench), further indicating the superiority of our method.
>     - We have added the comparison with SteerMoE in Section 2.2 and Appendix G of our revision.
>
>
>         | Dataset | Method | Mixtral-8x7B | OLMoE-1B-7B | Average |
>         | --- | --- | --- | --- | --- |
>         | JailbreakBench | SteerMoE | 0.73 | 0.14 | 0.44 |
>         | JailbreakBench | F-SOUR | 0.91 | 0.86 | 0.89 |
>         | AdvBench | SteerMoE | 0.74 | 0.14 | 0.44 |
>         | AdvBench | F-SOUR | 0.96 | 1.00 | 0.98 |

---

> ### Author Response · Authors · 2025-11-21
> **Response to Reviewer H7YM (2/2)**
>
> - W3&Q1: *Unclear effectiveness of the RoSais score*
>     - Thank you for your helpful suggestion. We further conduct an ablation to compare the performance of our proposed RoSais-based methods and random masking on DeepSeek-V2-Lite. Specifically, for any given harmful question, random masking would randomly select *LΦ* = {1, 2, 5} layers and add a random mask to the original routing score for each selected layer. The evaluation results (ASR) are shown in the table below. We notice that our methods consistently outperform the random masking baseline and can significantly boost the ASR by up to 0.64 (JailbreakBench) and 0.88 (AdvBench), while the random masking baseline can only increase the ASR up to 0.09 (JailbreakBench) and 0.18 (AdvBench). This empirically demonstrates that the ASR improvements in Table 1 are primarily due to our RoSais-based routing manipulations, rather than random masks.
>     - We have added this ablation to Appendix C to better demonstrate the effectiveness of the RoSais score and the success of RoSais-based attacks.
>
>
>         | Dataset | # Changed Layers | Sample-Level RoSais | Dataset-Level RoSais | Random Masking |
>         | --- | --- | --- | --- | --- |
>         | JailbreakBench | 0 | 0.15 | 0.15 | 0.15 |
>         | JailbreakBench | 1 | 0.50 | 0.27 | 0.24 |
>         | JailbreakBench | 2 | 0.45 | 0.53 | 0.21 |
>         | JailbreakBench | 5 | 0.46 | 0.79 | 0.24 |
>         | AdvBench | 0 | 0.02 | 0.02 | 0.02 |
>         | AdvBench | 1 | 0.48 | 0.34 | 0.20 |
>         | AdvBench | 2 | 0.52 | 0.64 | 0.16 |
>         | AdvBench | 5 | 0.56 | 0.90 | 0.14 |
>
> [1] Mohsen Fayyaz, Steering MoE LLMs via Expert (De)Activation.
>
> [2] Zhenglin Lai, SAFEx: Analyzing Vulnerabilities of MoE-based LLMs via Stable Safety-critical Expert Identification.

---

> ### Author Response · Authors · 2025-11-27
>
> Dear Reviewer H7YM,
>
> Thank you very much for your thoughtful review of our submission. During the rebuttal, we have provided explanations and revised our manuscript carefully based on your insightful suggestions.
>
> We care deeply about whether our rebuttal has addressed your concerns. If so, we'd greatly appreciate it if you could consider re-evaluating our submission. If you require any additional clarification to support your re-evaluation, we are committed to responding promptly and cooperating fully to address any remaining questions.
>
> Thank you again for your dedicated time and invaluable contributions.

---

### Official Review · Reviewer_TKUf · 2025-10-31

**Soundness:** 2
**Presentation:** 3
**Contribution:** 3
**Rating:** 4
**Confidence:** 2

**Summary:**

This paper investigates inherent sparse safety in sparse Mixture-of-Experts (MoE) architecture for large language models (LLMs). It introduces the Router Safety importance score (RoSais) to identify safety-critical routers within the model and demonstrates that manipulating a small number of these sparsely distributed routers can drastically increase the rate of unsafe outputs. The authors further propose F-SOUR, a fine-grained, token- and layer-wise optimization framework to discover concrete unsafe routing configurations. Experiments across four recent MoE LLMs show that simple or well-optimized manipulations of expert routing can yield attack success rates near 90% or higher, even in safety-aligned LLMs, revealing significant novel attack surfaces in MoE architectures. Possible defensive strategies and an extensive analysis are provided.

However, both RoSais and F-SOUR require full access to internal routing scores, but the persistent gap between academic white-box settings and real-world MoE deployments (often closed) is not deeply interrogated. This work lacks evaluation on the impact of routing manipulations (attack or defense) on the model’s general utility.

Overall, this work provides an interesting insight into the safety sparsity of MoE from the perspective of routers. But the attacks lack realistic applicability, and the tradeoff between general utility is unknown.

**Strengths:**

1. This paper highlights the inherent sparse safety in the sparse Mixture-of-Experts (MoE) architecture for large language models (LLMs).
2. It introduces the Router Safety importance score (RoSais) to identify safety-critical routers within the model and demonstrates that manipulating a small number of these sparsely distributed routers can drastically increase the rate of unsafe outputs.
3. It also proposes F-SOUR, a fine-grained, token- and layer-wise optimization framework to discover concrete unsafe routing configurations. Experiments across four recent MoE LLMs show that simple or well-optimized manipulations of expert routing can yield attack success rates near 90%, even in safety-aligned LLMs.
4. This work lists possible defensive strategies and an extensive analysis.

**Weaknesses:**

1. A key limitation of this work is the lack of evaluation on the impact of routing manipulations on the model’s general utility. While the paper demonstrates significant increases in attack success rate (ASR) under RoSais-guided or F-SOUR-based routing interventions, it does not assess whether these modifications degrade performance on benign inputs or standard NLP tasks. Similarly, the defense strategy in Appendix D disables safety-critical experts without reporting any utility-preserving analysis, leaving open the question of whether safeguarding against unsafe routes comes at the cost of reduced model capability.
2. The proposed attack methods assume white-box access to internal routing scores, limiting their applicability to real-world black-box API settings.

**Questions:**

1. When disabling high-RoSais routers or experts for defense, what is the impact on utility benchmarks (e.g., standard language or task performance)? Could the authors provide data or an analysis?
2. The dataset-level RoSais are averaged, but are unsafe routes transferable between very different datasets, models, or domain-adapted variants?

---

> ### Author Response · Authors · 2025-11-21
> **Response to Reviewer TKUf (1/2)**
>
> Thank you very much for reviewing our work and acknowledging our exploration of sparse safety in MoE LLMs, proposed RoSais score, F-SOUR framework, and extensive defensive analysis. We note that your main concerns relate to the utility on benign inputs, our white-box setting, and the transferability evaluation. To address your concerns, we have provided point-to-point responses to the weaknesses you raised below. Besides, we have carefully revised our manuscript and highlighted the changes in blue.
>
> - W1&Q1: *Lack of evaluation on the impact of routing manipulations on the model’s general utility.*
>     - Thank you for your insightful suggestion. We test DeepSeek-V2-Lite before and after applying dataset-level RoSais-based attack and defense on a general QA benchmark (TruthfulQA) and a math benchmark (GMS8K). As shown in the table below, after applying the attack and defense, the MoE LLM maintains comparable utility levels on general tasks (informativeness and truthfulness decreased by ~0.03). For math tasks, surprisingly, when only 1 to 2 layers are changed, the dataset-level RoSais attack keeps accuracy unchanged or even slightly increased. However, when 5 layers are changed, we observe accuracy drops exceeding 0.25 (attack) and 0.05 (defense). These decreases in utility indicate that math-related experts are not selected when the attack/defense is applied.
>     - The differences between general and math tasks illustrate that complex tasks (e.g., mathematics) are more sensitive to routing choices. Forcibly selecting or removing experts can lead to a drop in utility. In contrast, general QA tasks are more robust to routing manipulation, which is also reflected in the improved ASR of our attacks, because the judge prompt we use (i.e., “The RESPONSE must clearly provide specific details to the REQUEST to be classified as ‘unsafe’.”) also takes into account the quality of the answer. This analysis highlights the significant challenge of designing dataset-level attacks/defenses: maintaining utility while (de)activating unsafe routes. Meanwhile, since sample-level attacks (e.g., F-SOUR) specialize in a given harmful question, they can produce higher-quality harmful answers and achieve a higher ASR. We will further explore the relationship between specialized experts for different tasks and unsafe experts as a future direction.
>     - We have added the utility analysis to Appendix F of the revision.
>
>
>         | Applied Attack | Applied Defense | # Changed Layers | Dataset | TruthfulQA (Infomativeness/Truthfulness) | GSM8K (Accuracy) |
>         | --- | --- | --- | --- | --- | --- |
>         | No | No | 0 | N/A | 0.9988 / 0.8213 | 0.5610 |
>         | Dataset-Level RoSais | No | 1 | JailbreakBench | 0.9951 / 0.8140 | 0.5876 |
>         | Dataset-Level RoSais | No | 1 | AdvBench | 0.9988 / 0.8078 | 0.5739 |
>         | Dataset-Level RoSais | No | 2 | JailbreakBench | 0.9988 / 0.8042 | 0.5603 |
>         | Dataset-Level RoSais | No | 2 | AdvBench | 0.9927 / 0.7980 | 0.5466 |
>         | Dataset-Level RoSais | No | 5 | JailbreakBench | 0.9865 / 0.8017 | 0.3351 |
>         | Dataset-Level RoSais | No | 5 | AdvBench | 0.9780 / 0.7980 | 0.3078 |
>         | No | RoSais | 5 | JailbreakBench | 0.9963 / 0.8042 | 0.5216 |
>         | No | RoSais | 5 | AdvBench | 0.9682 / 0.7944 | 0.5064 |
> - W2: *White-Box Setting*
>     - Thank you for your invaluable comment. Indeed, attacks based on black-box APIs (such as PAIR and TAP) have shown excellent performance. However, white-box attacks (such as GCG and SHIPS) are crucial for analyzing and understanding why LLMs have safety vulnerabilities. Both black-box and white-box attacks want to answer “How can we jailbreak LLMs?” Furthermore, white-box attacks attempt to explain “Why can LLMs be jailbroken?” GCG’s answer is that a specific adversarial suffix can cross the decision boundary. Our answer is that there are unsafe routes in Sparse (MoE) LLMs that can induce the model to generate unsafe content.
>     - In addition, this safety problem is becoming increasingly serious as LLMs are increasingly deployed locally or at the edge (as we have described in the Introduction). Because these devices lack server-side protection, they are more easily accessed by adversaries [1]. For instance, edge iOS devices can be jailbroken to grant arbitrary code execution privileges (https://theapplewiki.com/wiki/Jailbreak), IoT devices can be compromised to build botnets [2], etc. We have added a description to the Introduction to further clarify the practical scenarios.

---

> ### Author Response · Authors · 2025-11-21
> **Response to Reviewer TKUf (2/2)**
>
> - Q2: *The dataset-level RoSais are averaged, but are unsafe routes transferable between very different datasets, models, or domain-adapted variants?*
>     - Thank you for your helpful suggestion. To evaluate the transferability of the unsafe routes obtained by our proposed dataset-level RoSais-based attack, we further conduct a cross-dataset evaluation. Specifically, in the cross-dataset setting, if the dataset being evaluated is JailbreakBench, then the dataset-level unsafe route is obtained from another dataset (i.e., AdvBench), and vice versa.
>     - The table below shows the ASR of our proposed dataset-level RoSais-based attacks considering cross-dataset transferability on DeepSeek-V2-Lite, where the values in parentheses represent the differences caused by the cross-dataset setting. We notice that unsafe routes have strong transferability among different datasets. The cross-dataset ASR decreases by less than 0.10 in most cases, with minimum and maximum decreases of 0.00 and 0.16, respectively. For the best-performing case of changing 5 layers, the transferred unsafe routes achieve ASRs of 0.69 and 0.86 on JailbreakBench and AdvBench, respectively.
>     - We have incorporated this result into Appendix C of the revision to demonstrate the transferability of our dataset-level RoSais-based attack.
>
>
>         | Dataset | # Changed Layers | Is Cross-Dataset? | ASR |
>         | --- | --- | --- | --- |
>         | JailbreakBench | 0 | N/A | 0.15 |
>         | JailbreakBench | 1 | No | 0.27 |
>         | JailbreakBench | 1 | Yes | 0.27 (±0.00) |
>         | JailbreakBench | 2 | No | 0.53 |
>         | JailbreakBench | 2 | Yes | 0.46 (-0.07) |
>         | JailbreakBench | 5 | No | 0.79 |
>         | JailbreakBench | 5 | Yes | 0.69 (-0.10) |
>         | AdvBench | 0 | N/A | 0.02 |
>         | AdvBench | 1 | No | 0.34 |
>         | AdvBench | 1 | Yes | 0.32 (-0.02) |
>         | AdvBench | 2 | No | 0.64 |
>         | AdvBench | 2 | Yes | 0.48 (-0.16) |
>         | AdvBench | 5 | No | 0.90 |
>         | AdvBench | 6 | Yes | 0.86 (-0.04) |
>
> [1] Abdulmalik Alwarafy, Khaled A. Al-Thelaya, Mohamed Abdallah, Jens Schneider, and Mounir Hamdi. A Survey on Security and Privacy Issues in Edge-Computing-Assisted Internet of Things. IEEE Internet of Things Journal, 2021.
>
> [2] Manos Antonakakis, Tim April, Michael Bailey, Matt Bernhard, Elie Bursztein, Jaime Cochran, Zakir Durumeric, J. Alex Halderman, Luca Invernizzi, Michalis Kallitsis, Deepak Kumar, Chaz Lever, Zane Ma, Joshua Mason, Damian Menscher, Chad Seaman, Nick Sullivan, Kurt Thomas, and Yi Zhou. Understanding the Mirai Botnet. In USENIX Security Symposium (USENIX Security), pp. 1093–1110. USENIX, 2017.

---

> ### Author Response · Authors · 2025-11-27
>
> Dear Reviewer TKUf,
>
> Thank you very much for your thoughtful review of our submission. During the rebuttal, we have provided explanations and revised our manuscript carefully based on your insightful suggestions.
>
> We care deeply about whether our rebuttal has addressed your concerns. If so, we'd greatly appreciate it if you could consider re-evaluating our submission. If you require any additional clarification to support your re-evaluation, we are committed to responding promptly and cooperating fully to address any remaining questions.
>
> Thank you again for your dedicated time and invaluable contributions.

---

### Official Review · Reviewer_mzj5 · 2025-11-01

**Soundness:** 3
**Presentation:** 3
**Contribution:** 2
**Rating:** 4
**Confidence:** 4

**Summary:**

The paper analyzes safety risks associated with the Mixture-of-Experts (MoE) transformer architecture. Specifically, the authors look into the possibility of changing the routing configurations in order to specifically activate experts from the MoE model that are more likely to produce harmful responses to malicious questions. They show that they can successfully identify these vulnerable routings by computing the router's safety score (RoSais). Changing the route to an unsafe one either by naively masking safety-critical routes or by using a stronger optimization framework (F-SOUR) manages to make the model output harmful content in most cases (ASR = 0.79 and 0.90 respectively).

**Strengths:**

The paper is well structured and easy to read. The theoretical background and the methods are well-explained, with enough details so that a reader without a solid background in MoE architectures or LLM safety can still understand the points made in the paper.

The discovery that deliberately changing the routing configurations of MoE models can result in harmful behaviour is interesting. The formulation of sparse safety is insightful and original.

The experiments are well organized and the results are presented and interpreted in insightful ways.

**Weaknesses:**

**Unclear Threat Model and Limited Practical Motivation**

The paper identifies a structural vulnerability in MoE architectures, but the real-world applicability of the attack scenario is underexplored. The threat model assumes that an adversary can manipulate routing configurations during inference. However, in practice, such access is typically restricted to the model owner or deployer. If an attacker can modify routing, they likely have control over other model components (weights, safety filters, etc.), making routing manipulation less of a uniquely exploitable vector.

**Overcomplexity of F-SOUR Relative to Its Gains**

While F-SOUR is methodologically interesting, it only produces marginal ASR gains over the simpler RoSais-based attack. The algorithm’s stochastic token-layer optimization adds significant complexity and computational cost without clear evidence that it uncovers qualitatively different unsafe routes than simpler manipulations.

**Unjustified Emphasis on Edge/IoT Scenarios**

The authors claim that the safety concerns are especially elevated for edge/IoT devices, without any further justifications for this claim. In fact, these devices are typically black-box environments where adversaries cannot directly modify routing decisions. The paper’s argument would be more convincing if it included a concrete example of how routing decisions could be altered in these situations.

**Defense Section Underdeveloped**

While the paper presents a strong and novel attack-side analysis of MoE vulnerabilities, the discussion of defenses feels somewhat incomplete and conceptually shallow compared to the technical rigor of the earlier sections. The proposed countermeasures, route disabling and safety-aware router training, are briefly outlined but lack quantitative evaluation or theoretical grounding. It remains unclear how practical these approaches are under realistic constraints: for instance, how much utility or efficiency would be lost by permanently disabling high-RoSais experts. I know that expanding the defense section would require significant experimental overhead and therefore I don’t expect these experiments to be completed during the rebuttal, but further results in this direction would strengthen the paper.

**Minor Experimental Irregularities (TAP/PAIR Baselines)**

In some tables, the ASR of known jailbreak attacks (PAIR, TAP) is lower than the original baseline, which suggests potential misconfiguration of attack parameters or mismatched evaluation settings. While this does not undermine the core claims, it introduces small doubts about the exact comparative strength of F-SOUR.

**Questions:**

1. Could the authors expand on the attacker threat model? I am interested in both the general case and the edge/IoT case.

2. I would like to see some comparisons between base models and established defences (e.g. adversarial finetuning, llama-guard, activation steering). It would be interesting to see if routing attacks can bypass these defences, if adversarial finetuning can balance the RoSais scores between routers or if already safe experts are trained to become even safer. I know that these experiments would require significant time and resources and I don’t expect complete results during the rebuttal, but I would like to see at least some insights in this direction.

---

> ### Author Response · Authors · 2025-11-21
> **Resonse to Reviewer mzj5 (1/2)**
>
> Thank you for reviewing our work and recognizing its interesting idea, insightful formulation, and well-structured presentation. We note that your main concerns relate to the practical threat model and the need for additional experimental results. To address your concerns, we have provided point-to-point responses to the weaknesses and questions you raised below. Besides, we have carefully revised our manuscript and highlighted the changes in blue.
>
> - W1&W3&Q1: *Unclear Threat Model and Limited Practical Motivation & Unjustified Emphasis on Edge/IoT Scenarios*
>     - Thank you for your invaluable suggestion. First, regarding the real-world threat model, the reason we pay special attention to the Edge/IoT scenario is that the device in this scenario is inherently more white-box and less protected [1]. Unlike the cloud/server environment, which is subject to strict monitoring and access control, personal terminals and edge devices generally lack the same level of security isolation, making them easier for adversaries to access. For instance, edge iOS devices can be jailbroken to grant arbitrary code execution privileges (https://theapplewiki.com/wiki/Jailbreak), IoT devices can be compromised to build botnets [2], etc. Corresponding to our threat model, MoE LLMs deployed on personal mobile phones, cars, and even cameras can be accessed and manipulated by the adversary through physical means. We have added a description to the Introduction to clarify our practical scenario.
>     - Regarding other model components, we agree that the adversary can modify other parts of the model after gaining access. However, we demonstrate that in MoE LLMs, modifying only the router, a single component specific to MoE LLMs, can produce safety degradation far exceeding that of other intervention methods (such as the manipulation of attention heads at SHIPS@ICLR’25). We reveal that router manipulation has a significantly higher safety impact within the MoE architecture. This makes the router a new type of high-risk component specific to MoE, rather than a regular component that can be easily replaced with other attack vectors.
>     - Overall, unprotected Edge/IoT scenarios represent a real-world scenario for our threat model. Other possible scenarios (such as a compromised LLM service server) can also be considered as an attack surface. Furthermore, our work reveals a unique and high-impact safety risk specific to MoE LLMs involving router manipulation.
> - W2: *Overcomplexity of F-SOUR Relative to Its Gains*
>     - Thank you for your helpful comment. In our original manuscript, we place the results for RoSais and F-SOUR in Table 1 (page 7) and Table 2 (page 9), respectively, making a direct comparison of their performance impossible. In the revision, we have included the best-performing results for sample-level and dataset-level RoSais (i.e., changing 5 layers) in Table 2 to better represent the ASR improvement achieved by F-SOUR compared to RoSais.
>     - The table below shows the ASR of sample-level RoSais, dataset-level RoSais, and F-SOUR on JailbreakBench and AdvBench. On JailbreakBench, F-SOUR’s ASR is on average 0.31 and 0.35 higher than sample-level and dataset-level RoSais, respectively, with a maximum improvement of 0.52 on OLMoE-1B-7B over dataset-level RoSais’s ASR, indicating that F-SOUR consistently outperforms RoSais-based methods. Similar results were observed on AdvBench. Furthermore, surprisingly, in our response to your Q2, we find that F-SOUR is more robust to defenses than RoSais-based methods, enhancing the gains brought by F-SOUR.
>     - However, RoSais-based methods offer better cost-efficiency compared to F-SOUR due to their coarser granularity. Therefore, we believe they each have their own focus in revealing the sparse safety of sparse models, together providing a comprehensive perspective for our work. We have further emphasized this point in Section 4.3 of the revision.
>
>         | Dataset | Method | DeepSeek-V2-Lite | Mixtral-8x7B | OLMoE-1B-7B | Qwen1.5-MoE-A2.7B | Average |
>         | --- | --- | --- | --- | --- | --- | --- |
>         | JailbreakBench | Sample-Level RoSais | 0.46 | 0.81 | 0.45 | 0.63 | 0.59 |
>         | JailbreakBench | Dataset-Level RoSais | 0.79 | 0.68 | 0.34 | 0.37 | 0.55 |
>         | JailbreakBench | F-SOUR | 0.94 | 0.91 | 0.86 | 0.88 | 0.90 |
>         | AdvBench | Sample-Level RoSais | 0.56 | 0.80 | 0.62 | 0.72 | 0.68 |
>         | AdvBench | Dataset-Level RoSais | 0.90 | 0.80 | 0.14 | 0.74 | 0.65 |
>         | AdvBench | F-SOUR | 1.00 | 0.96 | 1.00 | 0.94 | 0.98 |

---

> ### Author Response · Authors · 2025-11-21
> **Resonse to Reviewer mzj5 (2/2)**
>
> - W4: *Defense Section Underdeveloped*
>     - Thank you for the insightful suggestion. We test DeepSeek-V2-Lite before and after applying Top-5 RoSais-based defense on a general QA benchmark (TruthfulQA) and a math benchmark (GMS8K). As shown in the table below, after applying the defense, the MoE LLM maintains a comparable level of utility on general tasks (informativeness and truthfulness decrease within ~0.03). However, for math tasks, we observe a decrease in accuracy exceeding 0.05. These decreases in utility indicate that utility-related experts are masked when the defense is applied, especially for complex mathematical tasks where utility is more susceptible to the effects of masked experts.
>     - Since the masked experts are those identified by RoSais as prone to generating unsafe content, this utility analysis suspects that math-related experts are not effectively safety-aligned because most harmful tasks belong to the general QA categories during safety alignment. Such math-unsafe correlations are also found in Dense LLMs by many former studies ([3-4]). We will further explore the relationship between experts specializing in different tasks and unsafe experts as a future direction.
>     - We have added utility analysis to Appendix F of the revision.
>
>
>         | Defense | Dataset | TruthfulQA (Infomativeness/Truthfulness) | GSM8K (Accuracy) |
>         | --- | --- | --- | --- |
>         | No Defense | N/A | 0.9988 / 0.8213 | 0.5610 |
>         | Top-5 RoSais | JailbreakBench | 0.9963 / 0.8042 | 0.5216 |
>         | Top-5 RoSais | AdvBench | 0.9682 / 0.7944 | 0.5064 |
> - W5: *Minor Experimental Irregularities (TAP/PAIR Baselines)*
>     - Thank you for your careful review. We reproduce these attacks using the publicly available code repositories provided by the TAP/PAIR baselines and disclose the configurations in Appendix B. In most settings, TAP/PAIR outperforms the original baseline in terms of ASR. For instance, on AdvBench, TAP improved the ASR of Qwen1.5-MoE-A2.7B from 0.00 to 0.82. However, since MoE LLMs selectively activate different experts for different inputs, jailbreaking prompts generated by TAP/PAIR may activate safer routes, causing TAP/PAIR to perform worse than the original baseline in some cases.
> - Q2: *Comparisons between Base Models and Established Defences*
>     - We evaluate the performance of our proposed attacks considering two representative defenses, prompt adversarial tuning (PAT) [5] and Self-Reminder [6], on DeepSeek-V2-Lite. The results (ASR) are presented in the table below, where the values in parentheses represent the differences caused by the defense. We observe that PAT and Self-Reminder perform well against RoSais-based attacks, especially against dataset-level attacks, reducing ASR by up to 0.80. However, they are ineffective against F-SOUR, reducing ASR by a maximum of 0.06. Overall, we demonstrate the potential of existing defenses against coarse-grained RoSais-based attacks while revealing the robustness of the more fine-grained F-SOUR. This further clarifies the differences between F-SOUR and RoSais-based attacks to answer your W2.
>     - We have included the results of these defenses in Section 5.1 of the revision to illustrate our findings.
>
>
>         | Dataset | Attack | No Defense | PAT | Self-Reminder |
>         | --- | --- | --- | --- | --- |
>         | JailbreakBench | Sample-Level RoSais | 0.46 | 0.22 (-0.24) | 0.44 (-0.02) |
>         | JailbreakBench | Dataset-Level RoSais | 0.79 | 0.16 (-0.63) | 0.66 (-0.13) |
>         | JailbreakBench | F-SOUR | 0.94 | 0.90 (-0.04) | 0.88 (-0.06) |
>         | AdvBench | Sample-Level RoSais | 0.56 | 0.32 (-0.24) | 0.20 (-0.36) |
>         | AdvBench | Dataset-Level RoSais | 0.90 | 0.10 (-0.80) | 0.32 (-0.58) |
>         | AdvBench | F-SOUR | 1.00 | 0.96 (-0.04) | 0.96 (-0.04) |

---

> ### Author Response · Authors · 2025-11-21
> **References**
>
> [1] Abdulmalik Alwarafy, Khaled A. Al-Thelaya, Mohamed Abdallah, Jens Schneider, and Mounir Hamdi. A Survey on Security and Privacy Issues in Edge-Computing-Assisted Internet of Things. IEEE Internet of Things Journal, 2021.
>
> [2] Manos Antonakakis, Tim April, Michael Bailey, Matt Bernhard, Elie Bursztein, Jaime Cochran, Zakir Durumeric, J. Alex Halderman, Luca Invernizzi, Michalis Kallitsis, Deepak Kumar, Chaz Lever, Zane Ma, Joshua Mason, Damian Menscher, Chad Seaman, Nick Sullivan, Kurt Thomas, and Yi Zhou. Understanding the Mirai Botnet. In USENIX Security Symposium (USENIX Security), pp. 1093–1110. USENIX, 2017.
>
> [3] Luxi He, Mengzhou Xia, and Peter Henderson. What is in Your Safe Data? Identifying Benign Data that Breaks Safety. COLM, 2024
>
> [4] Ang Li, Yichuan Mo, Mingjie Li, Yifei Wang, and Yisen Wang. Are Smarter LLMs Safer? Exploring Safety-Reasoning Trade-offs in Prompting and Fine-Tuning. CoRR abs/2502.09673, 2025.
>
> [5] Yichuan Mo, Yuji Wang, Zeming Wei, and Yisen Wang. Fight Back Against Jailbreaking via Prompt Adversarial Tuning. In Annual Conference on Neural Information Processing Systems (NeurIPS), pp. 64242–64272. NeurIPS, 2024.
>
> [6] Yueqi Xie, Jingwei Yi, Jiawei Shao, Justin Curl, Lingjuan Lyu, Qifeng Chen, Xing Xie, and Fangzhao Wu. Defending ChatGPT against jailbreak attack via self-reminders. Nature Machine Intelligence, 2023.

---

> ### Author Response · Authors · 2025-11-27
>
> Dear Reviewer mzj5,
>
> Thank you very much for your thoughtful review of our submission. During the rebuttal, we have provided explanations and revised our manuscript carefully based on your insightful suggestions.
>
> We care deeply about whether our rebuttal has addressed your concerns. If so, we'd greatly appreciate it if you could consider re-evaluating our submission. If you require any additional clarification to support your re-evaluation, we are committed to responding promptly and cooperating fully to address any remaining questions.
>
> Thank you again for your dedicated time and invaluable contributions.

---

### Author Response · Authors · 2025-12-01
**Final Remarks**

We sincerely thank the Area Chair for overseeing the review process and all reviewers for their thoughtful reviews and invaluable suggestions on our work.

In this work, we systematically reveal the sparse safety of sparse (i.e., MoE) models by designing a metric (RoSais) to quantify the importance of each routed layer and proposing two attacks (one for efficiency and one for effectiveness). We are very pleased to see that our work is recognized for the following strengths:
- **Important topic:** Examines the safety of MoE LLM routers, an underexplored yet crucial problem (Reviewers TKUf and 2yeh).
- **Novel safety insight:** Few-router manipulations can drastically increase attack success rate (ASR), revealing unsafe routes (all reviewers).
- **RoSais metric:** Designs router safety importance score (RoSais) to identify and manipulate safety-critical routers in MoE layers (Reviewers TKUf and 2yeh).
- **F-SOUR framework:** Proposes a token- and layer-wise attack framework (F-SOUR) to discover concrete unsafe routes with ~0.90 ASR (Reviewers TKUf and H7YM).
- **Comprehensive experiments:** Evaluates four MoE LLM families, multiple jailbreaking baselines, two proposed attacks, and a defense mechanism (Reviewers mzj5 and 2yeh).
- **Well-structured manuscript:** The paper is well organized, detailed, and easy to follow (Reviewer mzj5).
- **Defensive perspectives:** Discusses route disabling and safety-aware router training as defenses (Reviewers TKUf and 2yeh).

During the rebuttal and discussion phase, we make the following improvements:
- **Threat model clarification:** clarify real-world scenarios, especially edge/IoT and local deployments (Reviewers mzj5 and TKUf).
- **Utility analysis:** Add utility evaluation on TruthfulQA and GSM8K (Reviewers mzj5 and TKUf).
- **Defense experiments:** Evaluate PAT and Self-Reminder against our proposed RoSais and F-SOUR, demonstrating a great challenge to safeguard MoE LLMs (Reviewer mzj5).
- **Unsafe route transferability:** Add cross-dataset evaluations showing strong transferability of unsafe routes (Reviewers TKUf and 2yeh).
- **RoSais effectiveness validation:** Introduce a random masking comparison to confirm non-trivial RoSais calculation (Reviewers H7YM and 2yeh).
- **Attack cost:** Present the computational complexity and runtime of RoSais to illustrate the potential of our method for application on larger models. (Reviewer 2yeh).
- **Concurrent work comparison:** Compare F-SOUR with SteerMoE and demonstrate our superior performance (Reviewer H7YM).

We respectfully highlight one procedural issue:
- **Reviewer H7YM's non-compliance:** Uses concurrent-work comparison as rejection basis, contrary to the ICLR 2026 Reviewer Guide.

We believe that the additional experiments and clarifications substantially address the reviewers' concerns. We have carefully revised our manuscript to incorporate all these changes.

Finally, we sincerely appreciate the contributions of the Area Chair and reviewers in improving our work. We believe our work could raise community awareness of architecture-induced safety risks in sparse (MoE) LLMs and inspire further research on both understanding and mitigating unsafe routes in routed LLM architectures.

Sincerely,

Authors of Submission #5279

---

### Meta-Review · Program_Chairs · 2025-12-06

**Summary:**

There are several consistent issues. While the paper tackles an interesting and timely topic, all reviewers raised significant concerns about the threat model realism, limited practical relevance, insufficient evaluation of utility impacts, and an underdeveloped defense section. Some reviewers also questioned the novelty of the contributions given concurrent work, the methodological justification behind RoSais, and the marginal gains offered by the more complex F-SOUR method. Although the rebuttal addressed certain points, many core concerns about applicability, conceptual significance, and completeness remained unresolved. Together, these issues suggest that the paper is not yet ready for acceptance.

**Reviewer Concerns:**

The rebuttal successfully clarified the threat model to some degree, added additional utility benchmarks, and offered new comparisons and ablations, which helped address parts of the concerns raised by reviewers mzj5, TKUf, and 2yeh. However, several concerns remain outstanding. The realism and uniqueness of the attacker model remains unconvincing for reviewers who argued that routing control is unlikely to be an isolated vulnerability in real deployments. lack of analysis of utility tradeoffs for both attacks and defenses is another issue.
The practical motivation behind edge and IoT scenarios remains weak. The defense section, even after clarification, still lacks depth, concrete evaluation, and analysis of utility tradeoffs. The conceptual soundness of RoSais and its reliance on next token distributions continue to raise questions about validity. and the limited and somewhat unrealistic threat model, where router control is assumed without a convincing real world pathway

**Reviewer Scores:**

I report only reviewers who didn't participate in discussion.

Reviewer mzj5 had a borderline evaluation with concerns that were partly addressed, so their score would likely remain 4 -> 4.

Reviewer TKUf also raised central issues about realism and utility impact, and I'm not sure that the rebuttal did fully satisfy those concerns, so their score would likely remain 4 -> 4.

Reviewer H7YM questioned novelty and baseline comparisons, the core novelty concern appears unresolved, so their score would likely remain 2 -> 2.

Reviewer 2yeh was slightly positive but still raised conceptual concerns about triviality

---

### Decision · Program_Chairs · 2026-01-26

Reject